# Tracing the emergence of domesticated grapevine in Italy

Mariano Ucchesu[1]*, Sarah Ivorra[1], Vincent Bonhomme[1], Thierry Pastor[1], Biancamaria Aranguren[2‡], Gianluigi Bacchetta[3], Giovanna Bosi[4], Andrea Cardarelli[5], Anna Depalmas[6], Gianni de Zuccato[7], Assunta Florenzano[4], Juan Francisco Gibaja-Bao[8], Marta Mariotti Lippi[9], Niccolò Mazzucco[10], Anna Maria Mercuri[4], Mario Mineo[11], Miria Mori Secci[9], Renato Nisbet[12], Gianluca Pellacani[13], Paola Perazzi[14‡], Mauro Rottoli[15], Luciano Salzani[16], Marco Sarigu[3], Alessandro Usai[17], Laurent Bouby[1]

1 ISEM, University of Montpellier-CNRS-IRD-EPHE, Montpellier, France, 2 Soprintendenza Archeologia, Belle Arti e Paesaggio per Siena, Grosseto Arezzo, Siena, Italy, 3 Centro Conservazione Biodiversità (CCB), Dipartimento di Scienze della Vita e dell'Ambiente (DI-SVA), Università degli Studi di Cagliari, Cagliari, Italy, 4 Laboratorio di Palinologia e Paleobotanica, Dipartimento Scienze Vita, Università di Modena e Reggio Emilia, Modena, Italy, 5 Dipartimento di Scienze dell'Antichità, Sapienza-Università di Roma, Rome, Italy, 6 DUMAS - Dipartimento di Scienze Umanistiche e Sociali, Università di Sassari, Sassari, Italy, 7 Former Archaeologist at Soprintendenza Archeologia, Belle Arti e Paesaggio per le province di Verona, Rovigo e Vicenza, Italy, 8 Institució Milà i Fontanals. Centro Superior de Investigaciones Científicas, Barcelona, Spain, 9 Università degli Studi di Firenze, Dipartimento di Biologia (BIO), Firenze, Italy, 10 Università di Pisa, Dipartimento di Civiltà e Forme del Sapere, Pisa, Italy, 11 Museo delle Civiltà di Roma, Rome, Italy, 12 Former Lecturer at Dipartimento di Studi Umanistici, Università Ca' Foscari, Venezia, Italy, 13 Museo Civico di Modena, Largo Porta Sant'Agostino, Modena, Italia, 14 Soprintendenza Archeologia, Belle Arti e Paesaggio per la città metropolitana di Firenze e le province di Pistoia e Prato, Firenze, Italy, 15 Laboratorio di Archeobiologia, Musei Civici di Como, Como, Italy, 16 Già Soprintendenza per i beni archeologici del Veneto – nucleo operativo di Verona, Verona, Italy, 17 Soprintendenza Archeologia, Belle Arti e Paesaggio per la città metropolitana di Cagliari e le province di Oristano e Sud Sardegna, Cagliari, Italy

‡ Retried
* marianoucchesu@gmail.com

## Abstract

This study presents an extensive analysis of 1,768 well-preserved waterlogged archaeological grape pips covering approximately 7000 years of history. These samples originate from 25 Italian archaeological sites spanning from the Early Neolithic (6th millemmium BC) to the Medieval period (8th-14th centuries AD). Employing geometric morphometrics and linear discriminant analyses, we compared these archaeological grape pips with modern reference collections to differentiate between wild and domestic grape types. Additionally, we analysed phenotypic changes in grape pip length and shape over the studied period to the present day to highlight traits associated with domestication syndrome. During the Early Neolithic, no evidence of morphologically domesticated grapes was observed. Data from Early Bronze Age sites (ca. 2050–1850 BC) display the same trend observed for the Early Neolithic period. The Middle Bronze Age sites (ca. 1600–1300 BC) continue to exhibit a predominance of wild grape pips. However, a notable transition occurs at the end of the Late Bronze Age (ca. 1300–1100 BC), with the majority of grape pips classified as domestic, indicating the definitive establishment of cultivation practices and selection of domestic grape by these communities. In the Iron Age, grape pips from Etruscan sites dating to the 4th century BC are predominantly domestic, suggesting

**Data availability statement:** All relevant data are within the paper and its Supporting Information files.

**Funding:** M.U. received funding from the European Union Horizon 2020 research and innovation programme under the Marie Skłodowska-Curie grant agreement (No 101019563 –VITALY). L.B. and S.I. were supported by the ANR MICA project (grant agreement ANR-22-CE27-0026).

**Competing interests:** The authors have declared that no competing interests exist.

an advanced viticulture for this period. During the Roman period (1st-6th centuries AD), some sites exhibited a high presence of domestic grape pips and intermediate forms between wild and domestic morphotypes, suggesting introgression between local wild and domestic grape allowing the formation of new varieties. Finally, the Medieval period (8th-14th centuries AD) sites demonstrate a widespread prevalence of domestic grape pips across archaeological sites, indicating a reduction of intermediate forms between wild and domestic morphotypes and displaying morphometric characteristics entirely similar to modern domestic grape references. Overall, our study provides valuable insights into the evolution of grapevine cultivation in Italy, highlighting the gradual transition from wild to domesticated types over millennia.

## Introduction

*Vitis vinifera* subsp. *sativa* (DC.) Hegi (domesticated grapevine) represents an important economic resource worldwide for wine production and the consumption of its fresh and dried fruits. According to the International Organization of Vine and Wine (OIV), grape cultivation covers 7.4 million hectares, yielding 77.8 million tons of fresh grapes and 260 million hectolitres of wine globally (http://www.oiv.int, world statistics year 2023).

Morphological studies and genetic analyses have revealed that cultivated grapevines were derived from domestication of *Vitis vinifera* subsp. *sylvestris* (C.C. Gmel.) Hegi (wild grape) [1–3]. Domestication induced a significant shift in the reproductive biology of grapevine: while wild plants are dioecious, most of domestic varieties are hermaphroditic, enabling self-pollination and fruit bearing [4].

The process of plant domestication implies genetic and morphological changes [5, 6]. In grapes, morphological changes are visible in the seed, with wild individuals typically featuring smaller, rounded seed with shorter beaks, while domestic types exhibit larger seeds with more elongated beaks [7].

Archaeobotanical studies indicate that the first domestication probably took place in southwestern Asia between the 6th and 3rd millennia BC [8–10]. Some researchers suggest that the southern Caucasus area is the most likely places where domestication occurred [11,12]. Different genomic analyses, have hypothesized that parallel or secondary events during grape domestication could have occurred in regions around the Mediterranean basin [13,14]. Recent genetic studies conducted by Dong and colleagues revealed that two separate domestication events occurred simultaneously about 11,000 years ago in Western Asia and the Caucasus area [15].

Starting from the 4th millennium BC, an increasing number of grape remains (pips, pedicels) were found in archaeological sites located in the northern Levant and Near East [1]. This evidence suggests a potential correlation with the emergence of complex societies that may have contributed to the spread of viticulture [1]. A similar pattern is observed in Italy, where hierarchical forms of social and political organization developed between the Early Bronze Age and the beginning of the Recent Bronze Age (ca. 2200–1150 BC) [16]. In this context, archaeological excavations reveal a notable increase in grape remains during the Bronze Age, suggesting the potential onset of grapevine cultivation [17–19].

Different research has employed pip outline analysis study grape subspecies [20–25]. This technique utilizes elliptic Fourier transforms (EFT), which translate object outlines into shape descriptors (Fourier coefficients = EFDs) [26]. Subsequently, the shape descriptors are utilised as geometric morphometric features in multivariate analyses to distinguish the two subspecies

[20]. This method has proven highly effective in distinguishing between wild and domestic grape pips [17,22,24,25,27,28]. Applying this methodology, a study conducted on archaeological grape pips from the site of Agia Paraskevi in Greece revealed grape pips with morphological similarities to domestic varieties dating to the Middle Bronze Age (ca. 1900–1700 BC) and Late Bronze Age (ca. 1500–1100 BC) [25,27]. Similarly, in Italy, the archaeological site of Sa Osa in central western Sardinia yielded waterlogged grape pips from the Middle Bronze Age (1391–1131 Cal. BC) that showed intermediate forms between wild and domestic grape and, from the end of the Late Bronze Age (1276–1078 Cal. BC), had instead morphological similarities with domestic grapes [17].

A recent study applying geometric morphometric data in combination with archaeogenetic analysis of waterlogged grape pips from the Pertosa Cave in Campania (southern Italy) established the presence of domestic grapevine during the Middle Bronze Age (ca. 1450–1200 BC), revealing a parent/offspring relationship between a domestic/wild hybrid individual and a domestic clonal group [18].

In Italy, archaeobotanical studies have documented the discovery of grape pips in different archaeological sites since the Early Neolithic period [19]. Unfortunately, these data are mostly limited to qualitative description of materials, except for a few cases where geometric morphometric analyses were carried out [17,18,28–30].

Italy's position in the center of the western Mediterranean has, since the Early Neolithic, facilitated cultural and technological exchanges with various prehistoric and protohistoric communities throughout the Mediterranean, contributing to the spread of viticulture [8,13,18].

While, the origin of viticulture seems to be understood in the Near East and eastern Mediterranean areas, in contrast, in the western Mediterranean and in particular in Italy, there are insufficient data to enable understanding of this complex phenomenon. Many questions have yet to be answered, for example, we do not know exactly when viticulture started, and we do not know if the ability to cultivate grapevine originated from the transmission of grape varieties, knowledge and techniques by eastern Mediterranean civilizations or if they have experimented independently grape cultivation from local wild grape populations.

In this light, we employ geometric morphometric analysis to explore the domestication status of 1,768 waterlogged grape pips discovered at 25 Italian archaeological sites spanning from the Early Neolithic to the Medieval Age.

Our objectives also extend to analysing phenotypic changes in grape pips length and shape over the period from the sixth millennium to the present day to highlight traits associated with domestication syndrome, shedding light on the emergence of the domesticated grapevine in Italy.

## Materials and methods

### Modern grape reference collection

The modern reference material consisted of pips from wild individuals and cultivars originating from Western and Central Europe, the Mediterranean area and Southwest Asia (S1 Table). Modern cultivar pips were obtained from the French ampelographic collection (INRAE, Vassal-Montpellier Grapevine Biological Resources Center, Marseillan-Plage, France; https://eng-vassal.montpellier.hub.inrae.fr/).

Wild materials were obtained from the collection of the ISEM-CNRS University of Montpellier and consisted of accessions from France, Italy, Spain, Greece and the Caucasus area (S1 Table). Following previous work [20], 30 pips were randomly selected for each domestic and wild accession.

## Archaeological grape pips

The archaeological grape pips come from 25 archaeological sites from northern-central Italy, Sardinia and from Republic of San Marino, dated between the Early Neolithic (6th millennium BC) and the Medieval Age (8th-14th centuries AD) (Fig 1, S2 Table), most of which included in the BRAIN-Botanical Record of Archaeobotany Italian Network database [31].

For this study, we selected 1,768 items only preserved in waterlogged conditions, because they serve as excellent samples for morphological comparisons with modern materials, as they do not exhibit the typical deformations observed in charred seeds; To establish the exact chronology of grape pips from undated stratigraphic units of archaeological sites, radiocarbon analysis was carried out on two grape pips from different stratigraphic units of the Early Neolithic site of La Marmotta (Lazio) and one grape pip from well KK of Sa Osa site (Sardinia) (S2 Table).

## Geometric morphometric analyses

Modern and archaeological grape pips were photographed in dorsal and lateral views using a stereomicroscope (Olympus SZ-ET) coupled with a digital camera (Olympus DP21). Subsequently, all the images were transformed into black silhouettes. The analysis of pip outlines was conducted through the use of elliptic Fourier transforms (EFT), as detailed in previous studies [20,21]. The EFT turns (x; y) coordinates of the outline into "Fourier coefficients" further treated as multivariate variables [32].

First, 360 equidistant points were sampled along the curvilinear abscissa. Then, the outlines were normalised for size, rotation, position and the first point. Consistent with prior research [20,21], the six first harmonics were utilized to describe each view, amounting to 48 coefficients (four coefficients per harmonic, two views). This choice, based on six harmonics, strikes a balance between shape description accuracy (capturing more than 95% of the total harmonic power) (S1 Fig) and minimizing measurement errors, which tend to increase with harmonic rank. The outline analyses were conducted using Momocs v. 1.4.0 (https://github.com/MomX/Momocs/; [32] in the R environment (v. 4.0.0) [33]. Pip length was measured manually using ImageJ by one single operator [34].

## Statistical analyses

All statistical analyses were conducted within the R environment, using the packages Momocs [32] and MASS [35]. The 48 EFT coefficients were used as quantitative variables of shapes (S3 Table). Principal Component Analysis (PCA) was used to explore shape differences between archaeological samples. Linear Discriminant Analysis (LDA) was used to infer the status (wild or domestic) of archaeological pips, previously trained on a modern reference collection composed of wild accessions and grape cultivars. We used a modern collection of two balanced sets composed of 2,430 wild and 2,430 domestic randomly selected pips (for raw data see Bonhomme et al. 2021). Leave-one-out cross-validation shows that 95.7% of the modern pips are correctly classified according to their status, wild or domestic. This LDA was then used to infer the status of the archaeological pips. Assignations with a posterior probability <0.9 were filtered out and considered as not allocated.

We represent chronological changes in pip size (seed length) and shape (coordinates on the first component of the PCA - PC1) by comparing pips of different archaeological periods or mean dates using boxplots and scatterplots. Mean dates were calculated considering the chronological period to which the archaeological remains belong. In the case of the Middle Bronze Age, Roman Age and Medieval Age, we considered the mean of multiple archaeological sites. When comparing chronological changes in seed size and shape we take into

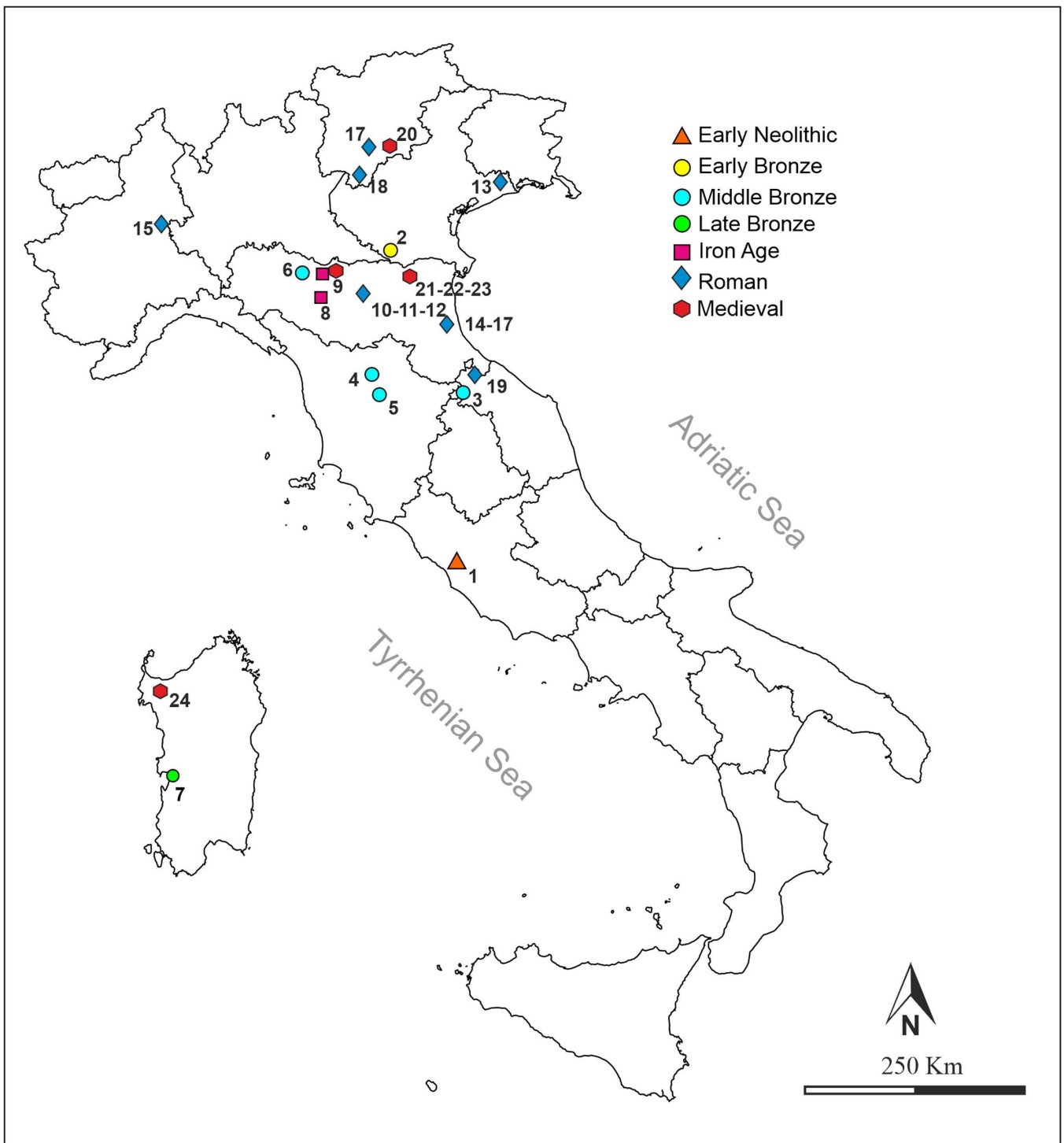

**Fig 1. Geographical distribution of the archaeological sites investigated:** (1) Anguillara Sabazia - La Marmotta, (2) Canàr di San Pietro in Polesine, (3) Pieve di Carpegna, (4) Gonfienti, (5) Firenze - San Lorenzo a Greve, (6) Noceto - Vasca votiva, (7) Cabras - Sa Osa, (8) San Polo d'Enza – Servirola, (9) Parma - Piazza Garibaldi (two contexts), (10) Modena - Area Novi Sad, (11) Modena - Viale Amendola, (12) Modena - Ex Cassa Risparmio, (13) Concordia Sagittaria - Porta Urbis, (14) Cervia - Rotatoria 71B, (15) Vercelli - Corso Prestinari, (16) Trento - Teatro sociale, (17) Classe - Condotto idrico, (18) Nago, (19) Domagnano, (20) Trento - Piazza Duomo, (21) Ferrara - Corso Porta Reno/via Vaspergolo, (22) Argenta - Via Vinarola/Aleotti, (23) Ferrara - Piazza Castello, (24) Sassari - Via Satta. (For more information on each individual archaeological site, see supplementary table S2 Table).

consideration the previously inferred status of the archaeological pips and compare them to the wild and domestic pips composing our reference collection.

## Results

### Inferring the wild/ domestic status of archaeological grape pips

LDA analysis carried out on the Early Neolithic site of Anguillara Sabazia - La Marmotta (1, Lazio) revealed that the majority of grape pips were identified as wild, while a small portion remained non allocated (Figs 2 and 3a, S4 Table). No domestic pip was identified.

Similarly, at the Early Bronze Age site of Canàr di San Pietro in Polesine (2, Veneto), most pips were classified as wild, with a small percentage non allocated (Figs 2 and 3a, S4 Table).

The results of the classification of the 142 pips found in four Middle Bronze Age sites located in Toscana, Marche and Emilia Romagna regions showed a prevalence of the wild morphotype. In this case, and for the first time, four pips were classified as domestic; with a smaller proportion were non allocated (Figs 2 and 3a, S4 Table).

From the single Late Bronze Age site of Sa Osa, located in Sardinia, the LDA revealed a near-equal distribution between domestic and no allocated pips, with a smaller proportion identified as wild (Figs 2 and 3a, S4 Table).

In the two Iron Age, Etruscan archaeological sites located in Emilia Romagna, the majority of pips were classified as domestic, with smaller proportion were classified as wild or non allocated (Figs 2 and 3b, S4 Table).

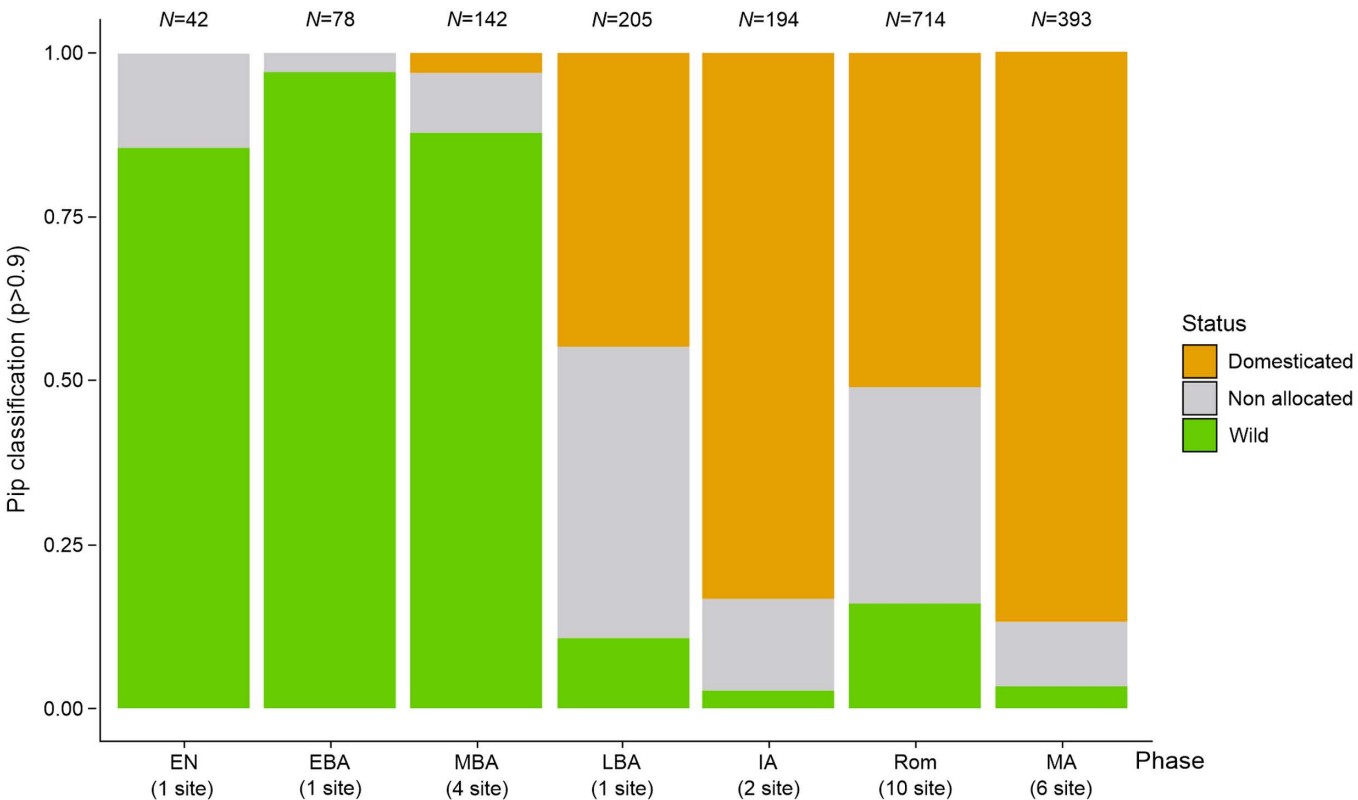

**Fig 2. Identification of the wild/ domestic status of the archaeological grape pips according to chronological periods.** All classifications with p < 0.9 are considered not significant (non allocated).

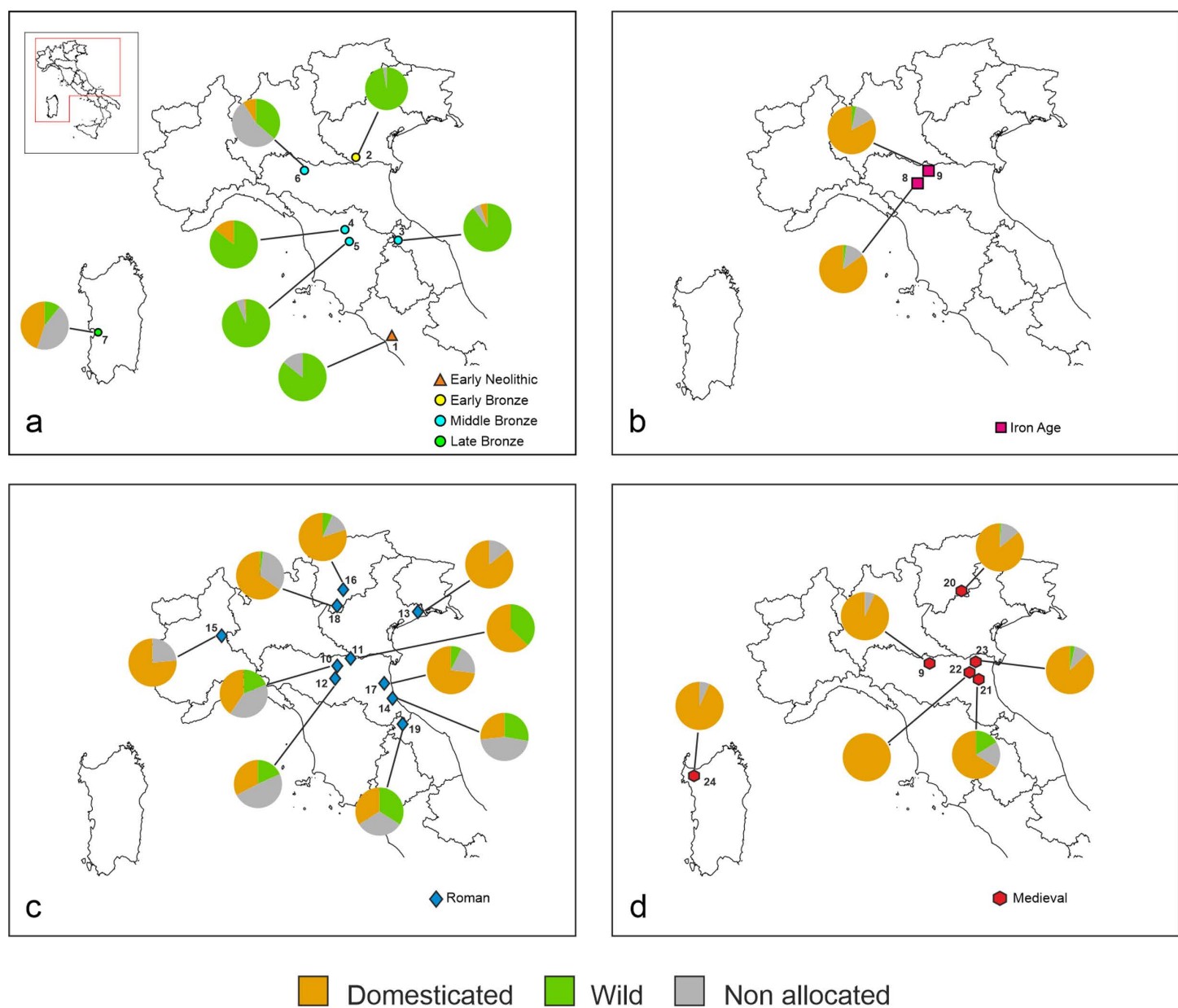

**Fig 3. Composition regarding status and spatial distribution of the archaeological grape assemblages arranged according to large time period (a, Neolithic to Late Bronze Age, b, Iron Age/Etruscan period, c, Roman period, d, Medieval times).** All classifications with p < 0.9 are considered not significant (non allocated).

The classification of grape pips from 10 archaeological Roman sites, located in regions of northern Italy and dated between the first and sixth centuries AD, exhibited a predominance of domestic pips (Fig 2, S4 Table). However, some Roman sites showed significant differences especially those located in southern Emilia Romagna. Sites such as Cervia and Domagnano showed an equal representation of wild and domestic pips, with a significant proportion were non allocated (Fig 3c, S4 Table).

For the Medieval period, grape pips from six archaeological sites located in Trentino-Alto Adige, Emilia Romagna and Sardinia, were mostly classified as domestic (Fig 2, S4 Table), with the sole exception of the Ferrara - Corso Porta Reno/via Vaspergolo site (21, Emilia

Romagna), which compared to other archaeological sites, showed a high percentage of grape pips allocated to the wild category (Fig 3d, S4 Table).

## Changes in grape pip size and shape over time

The observation of changes over time in the length of the archaeological grape pips revealed that it increased from the Neolithic and Early Bronze Age periods to the Middle-Late Bronze Age (ca. 1400–1200 BC) onwards (Figs 4 and 5, S5 Table). Grape pips allocated to the domestic morphotype are consistently longer than those allocated to the wild morphotype, although the difference between the median of the two types varies depending on the period. The length of the pips identified as domestic increased progressively from the Bronze Age to the present time. In particular, it increased between the Roman period and the Middle Ages, and then between the Middle Ages and the present day (Fig 5a and 5d, S5 Table). At the same time, the length of the pips classified as wild increased significantly from the Middle Bronze Age to the end of the Late Bronze Age, and then progressively until the Roman period (Fig 5b, S5 Table). At that time, the length of wild-type pips was clearly higher than that of current wild pips. Pip length seems to have decreased in the Middle Ages to come close to current figures (S5 Table).

Similarly, we conducted a comparative analysis of archaeological grape pips shape over time (Fig 6). The results showed that changes in shape are globally similar to changes in size (Fig 6). From the Bronze Age to the Middle Ages, pips classified as domestic gradually evolved into forms that were more and more typically domestic, i.e., more elongated, with a longer beak. Conversely, the varieties used to compose our reference database have a less typically domestic shape. At the same time, from the end of the Late Bronze Age to the Roman period, the shape of pips identified as wild also evolved towards a more domestic morphology, while

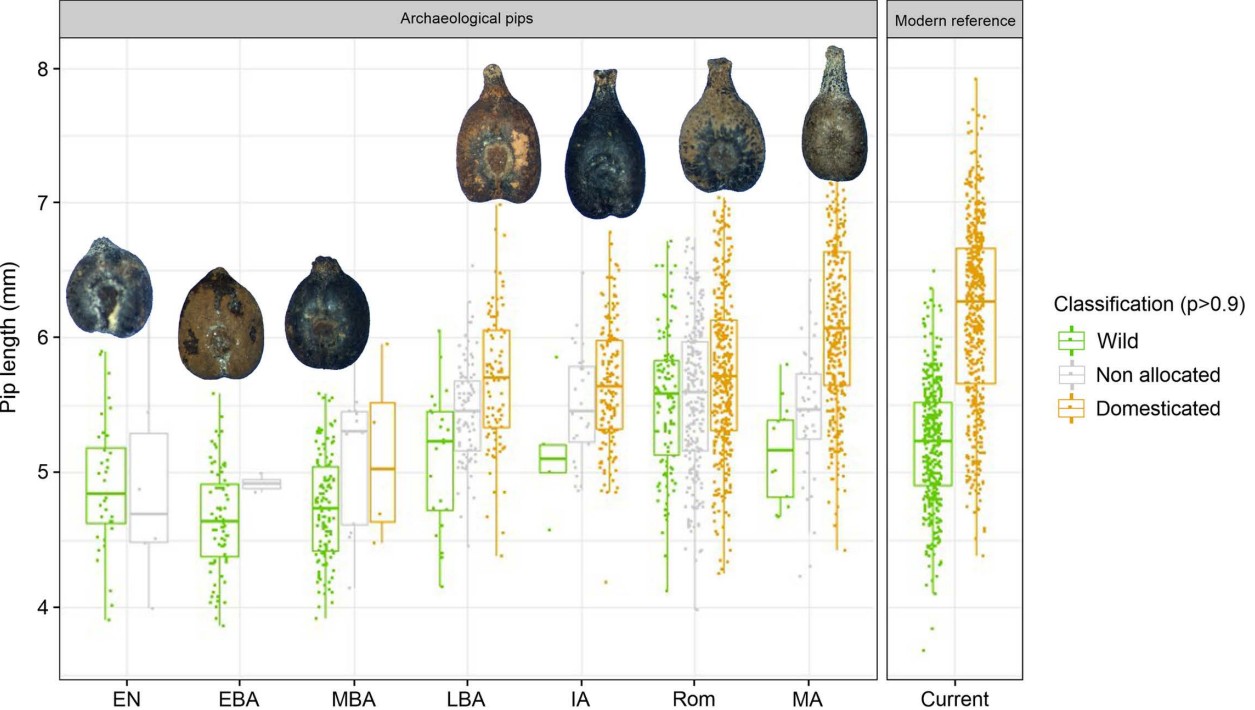

**Fig 4. Changes in grape pips length from the 25 studied archaeological sites, categorized into seven chronological periods, distinguishing seeds according to their classification into wild and domestic types, and comparison to current wild and domestic pips.**

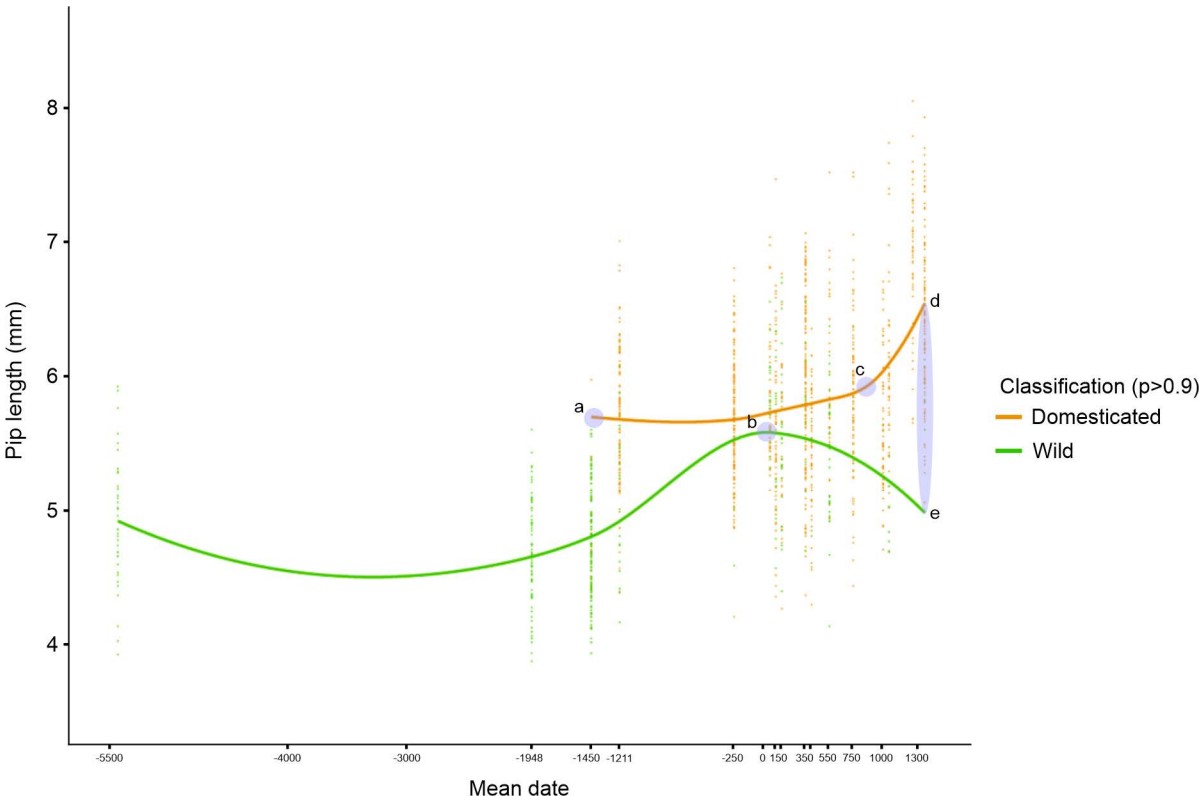

**Fig 5. Graphical representation of the trend in archaeological grape pip length changes over time using a loess smooth line (span = 0.75) reveals five key points: (a) the emergence of domestic grape forms; (b) the peak growth of wild grape pip length; (c) the peak growth of domestic grape pip length; (d-e) points of maximum divergence between the lengths of archaeological wild and domestic pips.**

maintaining significant differences from pips classified as domestic. In contrast, the wild-type pips from the Middle Ages show a morphology much closer to that of current wild pips. The gap between wild- and domestic-type pip morphology was never larger than during the Middle Ages.

## Discussion

The recent genetic studies performed on modern grape indicate that the origin of the great diversity of domestic grape varieties, present in the western Mediterranean, is the result of a pervasive introgressions between eastern domestic grapes and European wild populations [15,36]. Nonetheless, unraveling the origins of grapevines that emerged in western Mediterranean during ancient times remains a challenging issue, as comprehensive archaeogenetic and morphometric examinations of ancient grape pips have yet to be undertaken on a large scale.

In this study, we conducted an extensive analysis of 1,768 well-preserved waterlogged grape pips spanning approximately 7,000 years, originating from 25 Italian archaeological sites, ranging from the Early Neolithic (6th millennium BC) to the Medieval period (8th-14th centuries AD).

Utilising geometric morphometrics and linear discriminant analyses, we compared archaeological grape pips with modern reference collections to distinguish between wild and domestic morphotypes. Additionally, we analysed phenotypic changes in grape pip length and

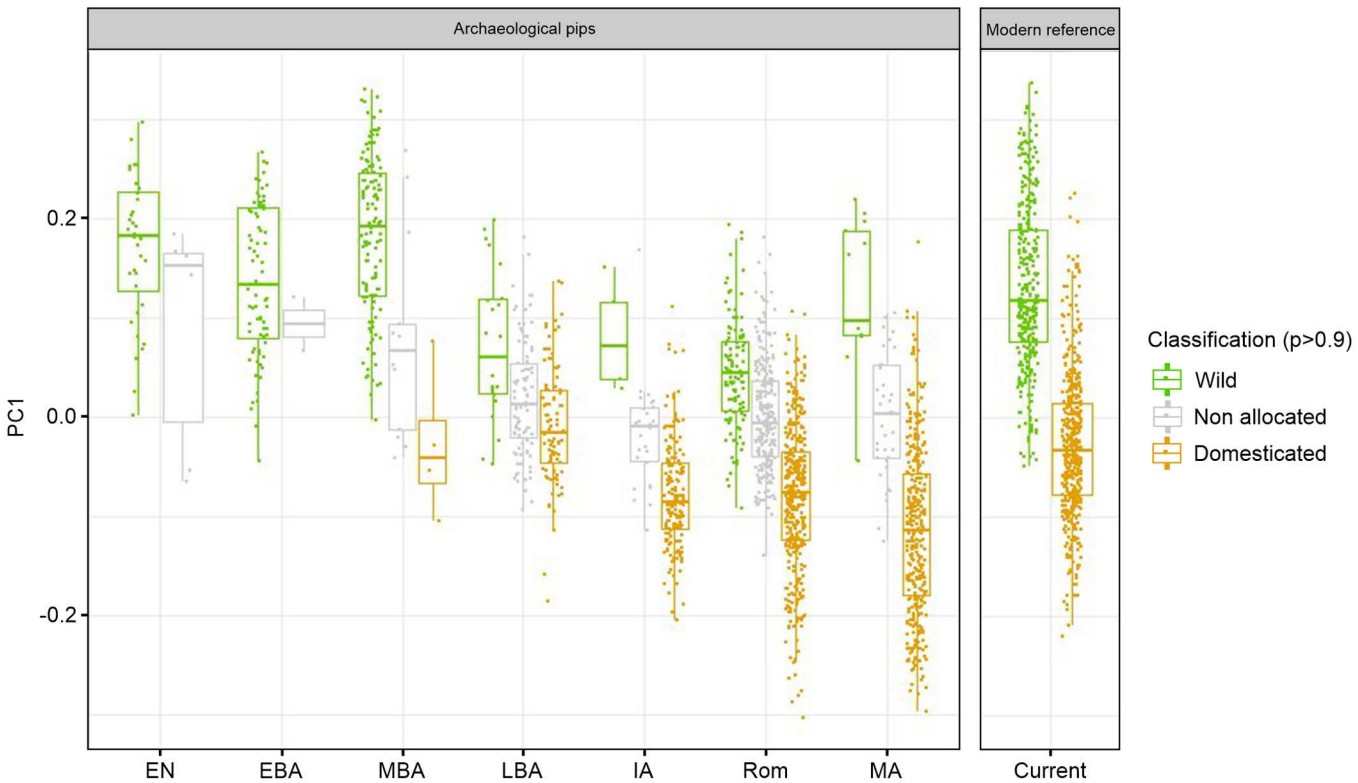

**Fig 6. Changes in the shape of the archaeological grape pips from the 25 studied archaeological sites, categorized into seven chronological periods, distinguishing seeds according to their classification into wild and domestic types, and comparison to current wild and domestic pip shapes.** The first axis of the PCA (PC1) performed on the EFT coefficients of the archaeological and modern pips was used as a shape variable.

shape over the period from the sixth millennium to the present day to highlight traits associated with domestication syndrome.

Our analyses revealed no evidence of morphologically domestic grapes during the Early Neolithic and Early Bronze Age periods; all grape pips analyzed exhibiting close similarities to modern wild grapes. These findings suggest that until at least the Early Bronze Age, human communities had not engaged in grapevine selection inducing domestication, with the presence of wild grape pips in these early archaeological contexts likely attributable to the gathering of berries from wild plants.

In reference to the Middle Bronze Age (ca. 1600–1300 BC), our analyses still highlight the presence of a high percentage of wild grape pips. Out of 142 grape pips from four archaeological sites, only four pips have been classified as domestic. Such a small percentage could correspond to the error of the model in classifying modern wild pips or to the possibility that some domestic traits naturally occur in grape pips of wild hermaphrodite populations [36] therefore cannot be considered significant. For this period, it is therefore not possible to establish a solid evidence of grapevine cultivation in northern Italy. Differently, a recent study that combined geometric morphometric and paleogenetic analyses conducted on 55 waterlogged grape pips dated to the Middle Bronze Age, found in Pertosa Cave (Campania) in southern Italy, revealed that 16% of the grape pips exhibited similarities with modern domestic grapes [18]. Similarly, a morphometric study conducted on waterlogged grape pips dated to the Middle Bronze Age, found in the site of Sa Osa in Sardinia, revealed the presence of grape pips that exhibited intermediate forms between wild and domestic grapes [17]. This could be

regarded as a proto-viticulture that could have been the result of external contacts between southern Italy and Sardinia and the Mycenaeans and Minoans [17,19]. East cultures may have transmitted techniques and introduced oriental domestic grape varieties [17,19]. Due to regular trade contacts between the eastern peoples and the protohistoric communities of southern Italy and Sardinia, it is possible to hypothesise that the eastern communities played a role in introducing early viticulture to these regions before it reached northern Italy. During the end of the Late Bronze Age (ca. 1300–1100 BC), a high percentage (45%) of grape pips from the single site of Sa Osa, in Sardinia, has been classified as domestic. These results, obtained from the grape pips found in the well KK and on the new batch of pips from the well N of Sa Osa, confirm the previous morphometric analyses carried out on the grape pips found in the well N of Sa Osa [17,27]. Furthermore, the comparative analysis of wild-type pips length and shape from Sa Osa revealed significant differences compared to wild pips from the earlier periods. The substantial changes in pip length and shape compared to the wild forms identified from Early Neolithic and Early Bronze Age sites suggest dynamic changes in grape cultivation practices, possibly influenced by the beginning of grape selection and cultivation carried out by the Late Bronze Age communities. Similar results have been found in Southern France, where wild-type pips from Iron Age sites – the period when the first domestic-type grapevines are encountered- differ in size and shape from both Neolithic and Bronze Age wild-type pips and from present-day wild pips [37]. Indeed, it has been highlighted that when wild grapes are cultivated, the shape and size of the pips and berries undergo changes, resulting in them being larger compared to counterparts growing in their natural environment [38].

Several hypotheses should be considered to explain the evidence of domestic grape pips identified at the site of Sa Osa. The domestic grapes could have been the result of a domestication process carried out by the Bronze Age communities of Sardinia, the introduction of fully domesticated grapevines from elsewhere, or the result of introgression between such imported domesticated grape varieties and wild grape plants naturally present in Sardinia. Recent paleogenetic analyses conducted on two grape seeds from well N at the Sa Osa site have revealed that they shared the same chlorotype found in modern cultivars from Armenia. This evidence, along with the results of our analyses, supports the hypothesis that allochthonous cultivars from the eastern Mediterranean were introduced to Sardinia during the Bronze Age [39].

The identification of wild-type, probably cultivated, grapes suggest a contribution of local wild grapevines to the emergence of viticulture in Sardinia. However, at the current stage of research, it is not possible to determine whether we should assume a secondary domestication process of grapes or merely introgression. This uncertainty is compounded by the fact that, during the Late Bronze Age, Sardinia was involved in extensive trade exchanges with Crete and Cyprus [40] which might have led to the import of domestic grape varieties and new knowledge about grapevine cultivation. However, the large quantities of domestic grape pips found in the wells of Sa Osa testify that during the end of the Late Bronze Age in Sardinia, grapevine cultivation was already well established. The evidence from Late Bronze Age of Sa Osa represents a single Italian context, and the absence in this study of grape pips from other Late Bronze Age sites, precludes generalizations to other Italian regions.

The interest in grape cultivation among the Late Bronze Age community in Sardinia may be linked to wine production as evidenced by recent chemical analyses of ceramics from Nuraghe Orroli and the Nuragic complex of Abini in Sardinia where organic residues associated with wine have been identified [41]. Similar evidence of wine production during the Bronze Age (15th-14th centuries BC) in Italy comes from Pilastri di Bondeno in the Po Valley (Ferrara, Emilia Romagna) and Canale Anfora (Aquileia, Udine), where chemical analyses of ceramic vessels revealed traces of wine [42]. However, wine production, documented in Bronze Age sites in northern Italy, cannot be conclusively linked to the cultivation of domestic

grape, as no grape pips were found, and since the wine can also be produced using wild grapes. This raises further questions about the cultivation of wild grapes for wine production, potentially marking the beginning of proto-viticulture. Evidence suggesting the use of wild grape, possibly implying also cultivation, during the Late Bronze Age comes from the site of Terramara Santa Rosa di Poviglio in Emilia Romagna [43]. In fact, based on the evidence that wild and domestic grapes have different types of flowers, a different pollen morphology can be observed in the two subspecies: hermaphrodite (domestic) and male (wild) flowers typically have trizonocolporate pollen, while female (wild) flowers have inaperturate pollen [44]. Pollen found at the Terramara Santa Rosa di Poviglio site displayed traits consistent with inaperturate pollen, suggesting the exploitation and potential cultivation of wild female grapes near the archaeological site [43].

During the Iron Age, the majority of grape pips recovered from two Etruscan sites (central Italy) dating to the 4th century BC were classified as domestic, with only a minor portion identified as wild. Our findings align with archaeological evidence and written sources affirming the Etruscan civilization's proficiency in grape cultivation and wine production [45–47]. Further evidence comes from the Cetamura site (Tuscany), where paleogenomic analyses performed on some grape pips dated to the 4th-3rd century BC, showed genetic affinities with wild populations and domestic Italian varieties [48]. Furthermore, geometric morphometric analyses conducted on Cetamura grape pips showed similarity to modern domestic grapes [49].

The production and trade of wine are well-documented from the last decades of the 7th century BC to the middle of the 5th century BC, as evidenced by the discovery in France, Spain, and North Africa of numerous Etruscan wine amphorae coming from Etruria, the modern southern Tuscany, northern Latium and s-w Umbria [45–47].

The analysis of grape pips length showed little difference between wild and domestic grape pips from the Late Bronze Age of Sardinia and northern Etruscan sites. Conversely, grape pip shape shows pronounced differences between domestic Late Bronze Age and Iron Age Etruscan specimens. This may indicate that at that time domestic grapes were morphologically much closer to modern varieties, potentially showing advanced selection and that during the Iron Age, domestic grape cultivation in Italy was well established.

The interest in grape cultivation during the Iron Age in Italy continued and intensified during the Roman period [50]. It is no coincidence that there is an increase in the discovery of large quantities of grape remains at Roman archaeological sites throughout the Italian peninsula [51].

The majority of grape pips from 10 sites of the Roman period (1st-6th centuries AD) were classified as domestic. A significant portion was non allocated, while a minor percentage was identified as wild.

While, in general, the Roman period exhibits a prevalence of domestic grapes over wild-type pips, some archaeological sites like Domagnano, Cervia, and Modena - Viale Amendola revealed significant variations, showcasing higher proportions of wild-type grapes compared to other sites. As suggested by the study of Bouby et al. 2013 [23], the co-presence of wild-type grape pips with domestic ones could be linked to selection pressures implemented by Roman viticulturists, who selected and cultivated wild grapes alongside domestic ones, possibly with the consequence of obtaining new varieties by introgression. In our case, size differences – and to a lesser extent shape differences- between wild-type and domestic-type pips are particularly reduced during Roman times, which reinforce the continuum between wild and domestic types, suggesting that all corresponded to cultivated vines, some of them being in a less advanced state of domestication.

This observation aligns with the previously formulated hypothesis for Roman sites in southern France by Bouby et al. (2013) [23], suggesting that the grapevines domestication

was a long and slow process. Consequently, it is plausible to suggest that the domestication process of grapes was still ongoing during the Roman period, even within the sites of peninsular Italy.

Finally, the Medieval period demonstrated a predominance of domesticated grape pips across most archaeological sites, reflecting the consolidation of grapevine domestication during this period. This is also confirmed by the analysis of grape pip length, which showed both wild and domestic types to be very similar to modern counterparts, suggesting that the experimental cultivation of wild grapes at the investigated sites, was no longer practiced during the Medieval period.

## Conclusion

Our study explored the evolutionary trajectory of grapevine cultivation in Italy through the analysis of archaeological grape pips spanning approximately 7,000 years. The findings shed light on the transition from wild grape gathering to intentional cultivation, marking key milestones in the history of viticulture.

During the Early Neolithic and Early Bronze Age periods, the absence of morphologically domestic grapes suggests a reliance on wild grape gathering, possibly with some experiment of proto-cultivation of wild grape.

Despite previous research showing the presence of domestic grapes in Middle Bronze Age sites such as Pertosa Cave in southern Italy and Sa Osa in Sardinia, our study of grape pips from several other sites do not reveal robust evidence for domestic grapes in Middle Bronze Age sites in northern Italy.

The Late Bronze Age represents a significant turning point, with a notable increase in domestic grape pips in the southern area, indicating advanced grape selection and cultivation practices. Further advancement in the presence of domesticated grape became evident in northern Italy during the Iron Age in Etruscan sites.

The Roman period saw the refinement of viticulture practices, with prevalence of domestic grape pips across most sites. However, intriguing variations in grape composition suggest an increased use of wild-type grapevines in viticulture and possibly introgression between local wild and domestic grapevines, leading to the development of new varieties. The Medieval period demonstrates a widespread prevalence of domestic grape pips that displayed morphometric characteristics entirely similar to modern domestic grape reference.

In conclusion, while this study spans several millennia, it presents a partial history of viticulture in Italy as not all regions have been investigated. Further research utilising multi-disciplinary approaches, combining geometric morphometric and paleogenomic analyses, will enhance our understanding of ancient grape cultivation and domestication, also considering its important socio-economic implications, enriching our knowledge of human-plant interactions throughout history.

## Supporting information

**S1 Table. List of modern reference of cultivars and wild used for the analyses.** (XLSX)

**S2 Table. List of the 25 Italian archaeological sites investigated.** ^ The site ID refers to the ID code assigned in the BRAIN-Botanical Record of Archaeobotany Italian Network database (https://brainplants.successoterra.net); *New C[14] data in this paper. Calibrated age ranges for each 14C date from IntCal20 using OxCal version 4.4.4 Atmospheric data from Reimer et al. 2020.
(XLSX)

**S1 Fig. Shape description accuracy.** For each view, a) cumulated amount of information brought by the first 12 harmonics based on a set of 100 randomly selected modern pips of the reference collection, b) reconstruction of pip shape for a randomly selected modern pip of the "Carignan" cultivar.
(PDF)

**S3 Table. The raw data of all the measurements conducted on archaeological pips.**
(XLSX)

**S4 Table. LDA seed classification (Threshold p ≥ 0.9).**
(XLSX)

**S5 Table. Mean and variance values for the 48 EFT coefficients and Pip length, for both the domesticated and wild types and for each period.**
(XLSX)

## Acknowledgments

We are grateful to the Centre de Ressources Biologiques de la Vigne, Domaine de Vassal-Montpellier (INRAE) (https://vassal.montpellier.hub.inrae.fr) which provided pips from modern cultivated varieties used in this study.

The authors thank the Soprintendenza Archeologia, Belle Arti e Paesaggio per la città metropolitana di Cagliari e le province di Oristano e Sud Sardegna for allowing us to analyse the seeds from Sa Osa and the Soprintendenza Archeologia, Belle Arti e Paesaggio per le province di Verona, Rovigo e Vicenza, and Giuseppe Zenezini for allowing us to analyse the seeds from Canar. Many thanks to, Gabriella Poggesi, Monica Salvini, and Andrea Pessina from Soprintendenza Archeologia, Belle Arti e Paesaggio per la città metropolitana di Firenze e le province di Pistoia e Prato for allowing us to analyse the seeds from S. Lorenzo a Greve and Gonfienti. Thanks to BRAIN network and database - https://brainplants.successoterra.net - are acknowledged for the archaeological sites mentioned in the paper.

The authors also thank Marta Mazzanti, Rossella Rinaldi (UNIMORE) and all students who worked with us on the sites cited in this work. Special thanks to Paola Bigi (Musei di Stato, Repubblica di San Marino), Mauro Cremaschi (Università degli Studi di Milano), Chiara Guarnieri, Donato Labate, Mirella Marini Calvani (già Soprintendenze Archeologia, Belle Arti e Paesaggio della Regione Emilia-Romagna), Roberto Macellari (già Musei Civici di Reggio Emilia), Daniela Rovina (già Soprintendenza Archeologia, Belle Arti e Paesaggio per le province di Sassari e Nuoro).

The research of the Marmotta site has been carried out in the collaboration agreement between the Museo delle Civiltà and the Spanish Scientific Research Council (centres in Barcelona IMF-CSIC and Rome EEHAR-CSIC). The authors would like to thank all the staff at the Museo delle Civiltà (curators, administrative staff, technicians, etc.) This research is part of the following research projects: 'AGER. Crescita agricola nell'Europa preistorica. Un approccio al cambio tecnologico, economico e sociale' project and the project 'Tools, Techniques and Specialists: the keys to understand the Mesolithic–Neolithic transition in Mediterranean Europe' A.M.M., A.F. and G.B. acknowledge the NRRP, Mission 4, Component 2 Investment 1.4 -Call for tender No.3138 of 16 December 2021 of MUR.

## Author contributions

**Conceptualization:** Mariano Ucchesu, Sarah Ivorra, Laurent Bouby.

**Data curation:** Mariano Ucchesu, Sarah Ivorra, Thierry Pastor, Laurent Bouby.

**Formal analysis:** Sarah Ivorra, Laurent Bouby.

**Funding acquisition:** Mariano Ucchesu.

**Investigation:** Mariano Ucchesu, Laurent Bouby.

**Methodology:** Mariano Ucchesu, Sarah Ivorra, Vincent Bonhomme, Thierry Pastor, Laurent Bouby.

**Resources:** Gianluigi Bacchetta, Giovanna Bosi, Andrea Cardarelli, Anna Depalmas, Gianni de Zuccato, Marta Mariotti Lippi, Miria Mori Secci, Renato Nisbet, Mauro Rottoli, Luciano Salzani, Alessandro Usai.

**Software:** Sarah Ivorra.

**Supervision:** Laurent Bouby.

**Validation:** Mariano Ucchesu, Sarah Ivorra, Biancamaria Aranguren, Giovanna Bosi, Andrea Cardarelli, Anna Depalmas, Gianni de Zuccato, Assunta Florenzano, Juan Francisco Gibaja-Bao, Marta Mariotti Lippi, Niccolò Mazzucco, Anna Maria Mercuri, Mario Mineo, Miria Mori Secci, Renato Nisbet, Gianluca Pellacani, Paola Perazzi, Mauro Rottoli, Luciano Salzani, Marco Sarigu, Alessandro Usai, Laurent Bouby.

**Visualization:** Vincent Bonhomme, Thierry Pastor, Biancamaria Aranguren, Gianluigi Bacchetta, Anna Depalmas, Gianni de Zuccato, Assunta Florenzano, Juan Francisco Gibaja-Bao, Marta Mariotti Lippi, Niccolò Mazzucco, Anna Maria Mercuri, Mario Mineo, Miria Mori Secci, Renato Nisbet, Gianluca Pellacani, Paola Perazzi, Mauro Rottoli, Luciano Salzani, Marco Sarigu, Alessandro Usai, Laurent Bouby.

**Writing – original draft:** Mariano Ucchesu, Sarah Ivorra, Laurent Bouby.

**Writing – review & editing:** Mariano Ucchesu, Vincent Bonhomme, Giovanna Bosi, Anna Depalmas, Assunta Florenzano, Marta Mariotti Lippi, Anna Maria Mercuri, Laurent Bouby.

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
