## [Decision Letter · Decision Letter 0]

23 Dec 2024

PONE-D-24-32533Tracing the Emergence of Domesticated Grapevine in ItalyPLOS ONE

Dear Dr. Ucchesu,

Thank you for submitting your manuscript to PLOS ONE. After careful consideration, we feel that it has merit but does not fully meet PLOS ONE’s publication criteria as it currently stands. Therefore, we invite you to submit a revised version of the manuscript that addresses the points raised during the review process.

We look forward to receiving your revised manuscript.

Kind regards,

Iris Groman-Yaroslavski, Ph.D

Academic Editor

PLOS ONE

Journal Requirements:

2. In your manuscript, please provide additional information regarding the specimens used in your study. Ensure that you have reported human remain specimen numbers and complete repository information, including museum name and geographic location. 

For more information on PLOS ONE's requirements for paleontology and archeology research, see https://journals.plos.org/plosone/s/submission-guidelines#loc-paleontology-and-archaeology-research.

M.U. received funding from the European Union Horizon 2020 research and innovation programme under the Marie Skłodowska-Curie grant agreement (No 101019563 –VITALY). L. Bouby and S. Ivorra were supported by the ANR MICA project (grat agreement ANR-22- CE27-0026). We are grateful to the Centre de Ressources Biologiques de la Vigne, Domaine de Vassal-Montpellier (INRAE) (https://vassal.montpellier.hub.inrae.fr) which provided pips from modern cultivated varieties used in this study. The authors thank the Soprintendenza Archeologia, Belle Arti e Paesaggio per la città metropolitana di Cagliari e le province di Oristano e Sud Sardegna for allowing us to analyse the seeds from Sa Osa and the Soprintendenza Archeologia, Belle Arti e Paesaggio per le province di Verona, Rovigo e Vicenza, and Giuseppe Zenezini for allowing us to analyse the seeds from Canar. Many thanks to, Gabriella Poggesi, Monica Salvini, and Andrea Pessina from Soprintendenza Archeologia, Belle Arti e Paesaggio per la città metropolitana di Firenze e le province di Pistoia e Prato for allowing us to analyse the seeds from S. Lorenzo a Greve and Gonfienti. Thanks to BRAIN network and database - https://brainplants.successoterra.net - are acknowledged for the archaeological sites mentioned in the paper. The authors also thank Marta Mazzanti, Rossella Rinaldi (UNIMORE) and all students who worked with us on the sites cited in this work. Special thanks to Paola Bigi (Musei di Stato, Repubblica di San Marino), Mauro Cremaschi (Università degli Studi di Milano), Chiara Guarnieri, Donato Labate, Mirella Marini Calvani (già Soprintendenze Archeologia, Belle Arti e Paesaggio della Regione Emilia-Romagna), Roberto Macellari (già Musei Civici di Reggio Emilia), Daniela Rovina (già Soprintendenza Archeologia, Belle Arti e Paesaggio per le province di Sassari e Nuoro). The research of the Marmotta site has been carried out in the collaboration agreement between the Museo delle Civiltà and the Spanish Scientific Research Council (centres in Barcelona IMF515 CSIC and Rome EEHAR-CSIC). The authors would like to thank all the staff at the Museo delle Civiltà (curators, administrative staff, technicians, etc.) This research is part of the following research projects: ‘AGER. Crescita agricola nell'Europa preistorica. Un approccio al cambio tecnologico, economico e sociale’ project (PGR18BQHM7) funded by the Italian Ministry for Education, Universities and Research within the ‘Rita Levi Montalcini’ program; and the project ‘Tools, Techniques and Specialists: the keys to understand the Mesolithic– Neolithic transition in Mediterranean Europe’ (PID2020-112513RB-I00) funded by MCIN/AEI/ 10.13039/501100011033. A.M.M., A.F. and G.B. acknowledge the NRRP, Mission 4, Component 2 Investment 1.4 -Call for tender No.3138 of 16 December 2021, rectified by Decree 363 n.3175 of 18 December 2021 of MUR funded by EU– NextGenerationEU. Project code CN_00000033, Concession Decree No. 1034 of 17 June 2022 adopted by MUR, CUP E93C22001090001, Project title “National Biodiversity Future Center– NBFC”

M.U. received funding from the European Union Horizon 2020 research and innovation programme under the Marie Skłodowska-Curie grant agreement (No 101019563 –VITALY). 

5. We note that you have indicated that there are restrictions to data sharing for this study. PLOS only allows data to be available upon request if there are legal or ethical restrictions on sharing data publicly. For more information on unacceptable data access restrictions, please see http://journals.plos.org/plosone/s/data-availability#loc-unacceptable-data-access-restrictions. 

6. We note that Figures 1 and 3 in your submission contain [map/satellite] images which may be copyrighted. All PLOS content is published under the Creative Commons Attribution License (CC BY 4.0), which means that the manuscript, images, and Supporting Information files will be freely available online, and any third party is permitted to access, download, copy, distribute, and use these materials in any way, even commercially, with proper attribution. For these reasons, we cannot publish previously copyrighted maps or satellite images created using proprietary data, such as Google software (Google Maps, Street View, and Earth). For more information, see our copyright guidelines: http://journals.plos.org/plosone/s/licenses-and-copyright.

a. You may seek permission from the original copyright holder of Figures 1 and 3 to publish the content specifically under the CC BY 4.0 license.  

Additional Editor Comments :

Dear Mariano Ucchesu,

The paper is very interesting and of high scientific value, however according to the review some more information and corrections is in needed. I also encourage you to go over the manuscripts in terms of language editing for final polishing and make sure references, figures and inferences are accurate. Please provide a detailed response letter elaborating all the corrections made.

Reviewers' comments:

Reviewer's Responses to Questions

**Comments to the Author**

1. Is the manuscript technically sound, and do the data support the conclusions?

Reviewer #1: Yes

Reviewer #2: Partly

2. Has the statistical analysis been performed appropriately and rigorously? 

Reviewer #1: Yes

Reviewer #2: I Don't Know

3. Have the authors made all data underlying the findings in their manuscript fully available?

Reviewer #1: No

Reviewer #2: No

4. Is the manuscript presented in an intelligible fashion and written in standard English?

Reviewer #1: Yes

Reviewer #2: Yes

5. Review Comments to the Author

Reviewer #1: The authors reported the morphometrics analysis of several waterlogged archaeological grape pips collected in Italian archaeological sites and covering 7000 years of history. Using geometric morphometrics and linear discriminant analyses, they suggested insights into the evolution of grapevine cultivation in Italy and the transition from wild to domesticated grapevines.

The manuscript reports interesting and original results and provides a potential contribution to understanding the diffusion and evolution of viticulture in Italy. These data should be supported in the future by genetic analyses that could suggest links with modern varieties cultivated in Italy and Southern Europe.

I suggest reporting the numerical results of the pips classification (wild, domesticate, non allocated) in the supplementary table SM2 next to the discovery sites. In this way the first part of the results (lines 229-260) could be lightened by many numerical data to make it less "boring" and similar to a list.

In addition, to meet the needs of open research, I would suggest that the authors report the raw data of all the measurements conducted on all the pips (in the supplementary materials or in some public repository), both the modern ones used to create the reference database and the archaeological ones.

Reviewer #2: 1. The method applied in the research, Fourier transforms, is widely used for morphometric identification of plants, and grape pips. It is important to add here in this paper a figure/ table to visualize the choice of the “six first harmonics, that were utilized here (although it is a continuation of the previously published method – Ref #20 and #30. Accordingly, the data of the measurements – the raw data for statistics, should be presented. Now, it is presented in figure 4, where individual measurements are unclear.

Next, the “48 EFT coefficients” – should be presented in the paper.

As for the results, means and covariances for the wild and domesticated types, as well, means and covariances for each period should be presented in the paper. This can be estimated from the training set. In this case, the data will help the readers to follow the authors and will allow us to evaluate the results, and to apply the results in future studies of other grape populations. Presenting the data will enlarge our understanding of the changes in the grape pip’s morphology with time and domestication.

Each method has its cautions and limitations, so it is critical to address this issue, and to discuss the pros and contra of applying the method on the results.

2. The reference list is incorrect. Page 2 [7] – Mangafa is not the source for the size and shape of wild vs cultivated pips. [13] – Dong et al. is the source for the two simultaneous domestication events 11,000 years ago in W Asia and Caucasus, which led to different routes and timing of dispersal and further introgressions of the primary cultivars. The reference [16] is dated to 2021, while [13] to 2023 – so the style of the sentence “However, the hypothesis of a second, independent, grape domestication event outside the original primary domestication center is still [sic! - reviewer] debated” - is confusing and misleading.

Page 2—Connection of the spread of viticulture with “emergence of complex societies”—please add more here for those who are not familiar with the development of societies in Italy. Add that in Italy, the first evidence of a hierarchical society was during the MB.

When citing only one single work in a sentence, such as “different research has employed pip outline analysis study grape subspecies [20] – the chosen reference should be a review, or please add “e.g.” before the single chosen work.

6. PLOS authors have the option to publish the peer review history of their article (what does this mean? ). If published, this will include your full peer review and any attached files.

**Do you want your identity to be public for this peer review?** For information about this choice, including consent withdrawal, please see our Privacy Policy .

Reviewer #1: No

Reviewer #2: No

---

## [Author Response · Author response to Decision Letter 1]

23 Jan 2025

Dear editor,

We are grateful for the helpful feedback by the reviewers that helped us to improve the quality of the manuscript. We carefully responded to all points.

Response to the academic editor:

1. Please ensure that your manuscript meets, including those for file naming.

Response 1

We followed the PLOS ONE's style requirements.

2. In your manuscript, please provide additional information regarding the specimens used in your study. Ensure that you have reported human remain specimen numbers and complete repository information, including museum name and geographic location.

Response 2

We added the following statement: “All necessary permits were obtained for the described study, which complied with all relevant regulations.”

We note that you have provided funding information that is not currently declared in your Funding Statement. However, funding information should not appear in the Acknowledgments section or other areas of your manuscript. We will only publish funding information present in the Funding Statement section of the online submission form. Please remove any funding-related text from the manuscript and let us know how you would like to update your Funding Statement. Currently, your Funding Statement reads as follows: M.U. received funding from the European Union Horizon 2020 research and innovation programme under the Marie Skłodowska-Curie grant agreement (No 101019563 –VITALY).

Response 3

We removed the funding information from the acknowledgments section; we would like the following statement to be included: “This research has received funding from the European Union’s Horizon 2020 research and innovation programme under the Marie Sklodowska-Curie grant agreement No 101019563. L. Bouby and S. Ivorra were supported by the ANR MICA project (grat agreement ANR-22-CE27-0026).”

5. We note that you have indicated that there are restrictions to data sharing for this study. PLOS only allows data to be available upon request if there are legal or ethical restrictions on sharing data publicly. For more information on unacceptable data access restrictions, please see http://journals.plos.org/plosone/s/data-availability#loc-unacceptable-data-access-restrictions.

b) If there are no restrictions, please upload the minimal anonymized data set necessary to replicate your study findings to a stable, public repository and provide us with the relevant URLs, DOIs, or accession numbers. For a list of recommended repositories, please see https://journals.plos.org/plosone/s/recommended-repositories. You also have the option of uploading the data as Supporting Information files, but we would recommend depositing data directly to a data repository if possible.

Response 5

All data are now publicly available (see further answers).

6. We note that Figures 1 and 3 in your submission contain [map/satellite] images which may be copyrighted. All PLOS content is published under the Creative Commons Attribution License (CC BY 4.0), which means that the manuscript, images, and Supporting Information files will be freely available online, and any third party is permitted to access, download, copy, distribute, and use these materials in any way, even commercially, with proper attribution. For these reasons, we cannot publish previously copyrighted maps or satellite images created using proprietary data, such as Google software (Google Maps, Street View, and Earth). For more information, see our copyright guidelines: http://journals.plos.org/plosone/s/licenses-and-copyright.

a. You may seek permission from the original copyright holder of Figures 1 and 3 to publish the content specifically under the CC BY 4.0 license.

“I request permission for the open-access journal PLOS ONE to publish XXX under the Creative Commons Attribution License (CCAL) CC BY 4.0 (http://creativecommons.org/licenses/by/4.0/). Please be aware that this license allows unrestricted use and distribution, even commercially, by third parties. Please reply and provide explicit written permission to publish XXX under a CC BY license and complete the attached form.” Please upload the completed Content Permission Form or other proof of granted permissions as an ""Other"" file with your submission.

Response 6

I have revised Figures 1 and 3. The current figures are the result of my personal work in vector graphics.

7. Please include captions for your Supporting Information files at the end of your manuscript, and update any in-text citations to match accordingly.

Response 7

We included the captions for our Supporting Information files at the end of our manuscript.

Reviewer's Responses to Questions

Reviewers' comments:

Reviewer #1: The authors reported the morphometrics analysis of several waterlogged archaeological grape pips collected in Italian archaeological sites and covering 7000 years of history. Using geometric morphometrics and linear discriminant analyses, they suggested insights into the evolution of grapevine cultivation in Italy and the transition from wild to domesticated grapevines.

The manuscript reports interesting and original results and provides a potential contribution to understanding the diffusion and evolution of viticulture in Italy. These data should be supported in the future by genetic analyses that could suggest links with modern varieties cultivated in Italy and Southern Europe.

I suggest reporting the numerical results of the pips classification (wild, domesticate, non allocated) in the supplementary table SM2 next to the discovery sites. In this way the first part of the results (lines 229-260) could be lightened by many numerical data to make it less "boring" and similar to a list.

Response

Following your suggestion, we have included all the seed classification results in the supporting file SM2, now renamed “S2A Table and S2B Table”. We also revised part of the manuscript by removing numerical data from the text.

In addition, to meet the needs of open research, I would suggest that the authors report the raw data of all the measurements conducted on all the pips (in the supplementary materials or in some public repository), both the modern ones used to create the reference database and the archaeological ones.

Response

We reported the raw data of all the measurements conducted on archaeological pips in the supplementary file “S3 Table” The full dataset related to the modern reference collection was already made available in Bonhomme et al. 2021.

Reviewer #2:

1. The method applied in the research, Fourier transforms, is widely used for morphometric identification of plants, and grape pips. It is important to add here in this paper a figure/ table to visualize the choice of the “six first harmonics, that were utilized here (although it is a continuation of the previously published method – Ref #20 and #30. Accordingly, the data of the measurements – the raw data for statistics, should be presented. Now, it is presented in figure 4, where individual measurements are unclear.

Response

All data related to the modern reference collection and archaeological pips are now fully available (see answer to previous comment). We have added as supplementary data a figure showing a) for each view, the cumulated amount of information brought by each of the first 12 harmonics for a set of 100 randomly selected modern seeds of the reference collection, and b) a reconstruction of seed shape a randomly selected pip of the “Carignan” modern cultivar based on the information brought by each harmonic (S1_fig.pdf).

Next, the “48 EFT coefficients” – should be presented in the paper. As for the results, means and covariances for the wild and domesticated types, as well, means and covariances for each period should be presented in the paper. This can be estimated from the training set. In this case, the data will help the readers to follow the authors and will allow us to evaluate the results, and to apply the results in future studies of other grape populations. Presenting the data will enlarge our understanding of the changes in the grape pip’s morphology with time and domestication. Each method has its cautions and limitations, so it is critical to address this issue, and to discuss the pros and contra of applying the method on the results.

Response

We added a table providing mean and variance values for the 48 EFT coefficients and Pip length, for both the domesticated and wild types and for each period (S4 Table).

2. The reference list is incorrect. Page 2 [7] – Mangafa is not the source for the size and shape of wild vs cultivated pips. [13] – Dong et al. is the source for the two simultaneous domestication events 11,000 years ago in W Asia and Caucasus, which led to different routes and timing of dispersal and further introgressions of the primary cultivars. The reference [16] is dated to 2021, while [13] to 2023 – so the style of the sentence “However, the hypothesis of a second, independent, grape domestication event outside the original primary domestication center is still [sic! - reviewer] debated” - is confusing and misleading.

Response 2.

Thank you for the suggestion. We have replaced the reference and revised the sentence to make it clearer.

Page 2—Connection of the spread of viticulture with “emergence of complex societies”—please add more here for those who are not familiar with the development of societies in Italy. Add that in Italy, the first evidence of a hierarchical society was during the MB.

Response

We have added a sentence to indicate the emergence of complex societies in Italy.

When citing only one single work in a sentence, such as “different research has employed pip outline analysis study grape subspecies [20] – the chosen reference should be a review, or please add “e.g.” before the single chosen work.

Response

We have added the relevant bibliographic references to the works that used the pip outline analysis study grape subspecies.

Peer review report for “Tracing the Emergence of Domesticated Grapevine in Italy”

What are the main claims of the paper and how significant are they for the discipline?

The paper’s focus is the question of the appearance of the domesticated form of grapes in Italy. This is the continuation of the research – see REF # 17-19. In a way, the presented research elaborates the method, and the data presented in Breglia, F., Bouby, L., Wales, N. et al. Disentangling the origins of viticulture in the western Mediterranean. Sci Rep 13, 17284 (2023). https://doi.org/10.1038/s41598-023-44445-4

Are the claims properly placed in the context of the previous literature? Have the authors treated the literature fairly?

The reference list is incorrect. Page 2 [7] – Mangafa is not the source for the size and shape of wild vs cultivated pips. [13] – Dong et al. is the source for the two simultaneous domestication events 11,000 years ago in W Asia and Caucasus, which led to different routes and timing of dispersal and further introgressions of the primary cultivars. The reference [16] is dated to 2021, while [13] to 2023 – so the style of the sentence “However, the hypothesis of a second, independent, grape domestication event outside the original primary domestication center is still [sic! - reviewer] debated” - is confusing and misleading. ETC. It must be updated and fixed.

Response

Already answered above.

Page 2—Connection of the spread of viticulture with “emergence of complex societies”—please add more here for those who are not familiar with the development of societies in Italy. Add that in Italy, the first evidence of a hierarchical society was during the MB.

Response

Already answered above.

When citing only one single work in a sentence, such as “different research has employed pip outline analysis study grape subspecies [20] – the chosen reference should be a review, or please add “e.g.” before the single chosen work.

Response

Already answered above.

Do the data and analyses fully support the claims? If not, what other evidence is required?

The method applied in the research, Fourier transforms, is widely used for morphometric identification of plants, and grape pips. It is a strong method for machine learning identification. Unfortunately, the method cannot be used for manual identification, but this will be, hopefully, addressed later.

Nevertheless, it is important to add here in this paper a figure/ table to visualize the choice of the “six first harmonics, that were utilized here (although it is a continuation of the previously published method – ref 20 and 30. Accordingly, the data of the measurements – the raw data for statistics, should be presented. Now, it is presented in figure 4, where individual measurements are unclear. Next, the “48 EFT coefficients” – should be presented in the paper.

Response

Already answered above.

As for the results, means and covariances for the wild and domesticated types, as well, means and covariances for each period should be presented in the paper. This can be estimated from the training set. In this case, the data will help the readers to follow the authors and will allow us to evaluate the results, and to apply the results in future studies of other grape populations. Presenting the data will enlarge our understanding of the changes in the grape pip’s morphology with time and domestication.

Each method has its cautions and limitations, so it is critical to address this issue, and to discuss the pros and contra of applying the method on the results.

Response

Already ans

---

## [Decision Letter · Decision Letter 1]

9 Feb 2025

PONE-D-24-32533R1Tracing the Emergence of Domesticated Grapevine in ItalyPLOS ONE

Dear Dr. Ucchesu,

Thank you for submitting your manuscript to PLOS ONE. After careful consideration, we feel that it has merit but does not fully meet PLOS ONE’s publication criteria as it currently stands. Therefore, we invite you to submit a revised version of the manuscript that addresses the points raised during the review process.

We look forward to receiving your revised manuscript.

Kind regards,

Iris Groman-Yaroslavski, Ph.D

Academic Editor

PLOS ONE

Journal Requirements:

Additional Editor Comments:

Please note the last remarks by reviewer 2.

Reviewers' comments:

Reviewer's Responses to Questions

**Comments to the Author**

1. If the authors have adequately addressed your comments raised in a previous round of review and you feel that this manuscript is now acceptable for publication, you may indicate that here to bypass the “Comments to the Author” section, enter your conflict of interest statement in the “Confidential to Editor” section, and submit your "Accept" recommendation.

Reviewer #2: All comments have been addressed

2. Is the manuscript technically sound, and do the data support the conclusions?

Reviewer #2: Yes

3. Has the statistical analysis been performed appropriately and rigorously? 

Reviewer #2: Yes

4. Have the authors made all data underlying the findings in their manuscript fully available?

Reviewer #2: Yes

5. Is the manuscript presented in an intelligible fashion and written in standard English?

Reviewer #2: Yes

6. Review Comments to the Author

Reviewer #2: **The paper presents important data that visualizes the non-linear evolution of wine domesticates in Italy. The raw data included now in the paper helps to follow the results and conclusions. I would suggest a minor change—in my opinion, a photo of grape pips from different stages of domestication would illustrate the process even better and significantly upgrade the visual appearance of the publication.**

7. PLOS authors have the option to publish the peer review history of their article (what does this mean? ). If published, this will include your full peer review and any attached files.

**Do you want your identity to be public for this peer review?** For information about this choice, including consent withdrawal, please see our Privacy Policy .

Reviewer #2: **Yes: ** Suembikya Frumin

---

## [Author Response · Author response to Decision Letter 2]

6 Mar 2025

Response to reviewer 2

Reviewers' comments:

Reviewer#2:

The paper presents important data that visualizes the non-linear evolution of wine domesticates in Italy. The raw data included now in the paper helps to follow the results and conclusions. I would suggest a minor change—in my opinion, a photo of grape pips from different stages of domestication would illustrate the process even better and significantly upgrade the visual appearance of the publication.

Response

Thank you for your suggestion. We have included the images of the archaeological grape seeds within Figure 4. Each seed is positioned in line with the statistical results related to seed length.

---

## [Editor Report · Decision Letter 2]

10 Mar 2025

Tracing the Emergence of Domesticated Grapevine in Italy

PONE-D-24-32533R2

Dear Dr. Mariano Ucchesu,

We’re pleased to inform you that your manuscript has been judged scientifically suitable for publication and will be formally accepted for publication once it meets all outstanding technical requirements.

Kind regards,

Iris Groman-Yaroslavski, Ph.D

Academic Editor

PLOS ONE
---

## [Editor Report · Acceptance letter]

PONE-D-24-32533R2

PLOS ONE

Dear Dr. Ucchesu,

I'm pleased to inform you that your manuscript has been deemed suitable for publication in PLOS ONE. Congratulations! Your manuscript is now being handed over to our production team.

Kind regards,

on behalf of

Dr. Iris Groman-Yaroslavski

Academic Editor

PLOS ONE